# Strategy of Coniferous Needle Biorefinery into Value-Added Products to Implement Circular Bioeconomy Concepts in Forestry Side Stream Utilization

**DOI:** 10.3390/molecules28207085

**Published:** 2023-10-14

**Authors:** Linards Klavins, Karolina Almonaitytė, Alvija Šalaševičienė, Alise Zommere, Kaspars Spalvis, Zane Vincevica-Gaile, Risto Korpinen, Maris Klavins

**Affiliations:** 1Department of Environmental Science, University of Latvia, Raina Blvd. 19, LV-1586 Riga, Latvia; alise.zommere@lu.lv (A.Z.); zane.gaile@lu.lv (Z.V.-G.); maris.klavins@lu.lv (M.K.); 2Food Institute, Kaunas University of Technology, Radvilenu Rd. 19, LT-50254 Kaunas, Lithuania; karolina.almonaityte@ktu.lt (K.A.); alvija.salaseviciene@ktu.lt (A.Š.); 3Biolat JSC, Rigas Str. 111, LV-2169 Salaspils, Latvia; kaspars.spalvis@biolat.lv; 4Biomass Fractionation Technologies, Production Systems, Natural Resources Institute Finland, Viikinkaari 9, FI-00790 Helsinki, Finland; risto.korpinen@luke.fi

**Keywords:** picea, pinus, biorefinery, extraction, coniferous, needles, valorization, bioeconomy, biorefinery, resin

## Abstract

Sustainable development goals require a reduction in the existing heavy reliance on fossil resources. Forestry can be considered a key resource for the bioeconomy, providing timber, energy, chemicals (including fine chemicals), and various other products. Besides the main product, timber, forestry generates significant amounts of different biomass side streams. Considering the unique and highly complex chemical composition of coniferous needle/greenery biomass, biorefinery strategies can be considered as prospective possibilities to address top segments of the bio-based value pyramid, addressing coniferous biomass side streams as a source of diverse chemical substances with applications as the replacement of fossil material-based chemicals, building blocks, food, and feed and applications as fine chemicals. This study reviews biorefinery methods for coniferous tree forestry biomass side streams, exploring the production of value-added products. Additionally, it discusses the potential for developing further biorefinery strategies to obtain products with enhanced value.

## 1. Introduction

The reduction in the heavy reliance on fossil fuels is essential to attain sustainable development goals. European Union (EU) policies concerning climate change mitigation require a strong reduction in fossil fuel use for energy production [1], to achieve climate-neutral development aims. As an alternative to a fossil material-based economy, the concept of bioeconomy can be considered, which supports social and technological transformation and innovation to replace fossil, non-renewable materials with renewable resources [2,3]. The goal of the bioeconomy is the “production of renewable biological resources and the conversion of these resources and waste streams into value-added products, such as food, feed, bio-based products and bioenergy” [4]. The progress of the bioeconomy can help replace over 90% of petroleum products with substances derived from renewable raw materials and replace one-third of fine chemicals, other materials, and half of the pharmaceuticals with biomass-based products by the year 2030 [5]. Technological innovation in biomass processing can supply industries with bio-based products that have a lower energy consumption, are environmentally friendly, and are less toxic, limiting the use of dangerous, especially persistent, chemicals and at the same time achieving high yields of target substances [6]. In a broader sense, the bioeconomy can offer economic benefits, production diversification, social and technological innovation in industries, new employment opportunities, as well as increased security in energy and industrial resource supply chains.

One challenge related to biomass conversion into different products is the complex composition of biomass. However, this complexity also provides an opportunity to obtain multiple products from a single batch of biomass, similar to different oil products (petroleum ether, gasoline, diesel, oil, etc.) obtained during petroleum refining from crude oil [7]. A widely accepted definition of biorefinery was given by the International Energy Agency as “the sustainable processing of biomass into a spectrum of marketable products and energy” [8]. The biorefinery concept covers different generations of biomass processing approaches and technologies that provide opportunities to transform different types of biomass resources (including those from agriculture, forestry, blue biotechnology, and others) into groups of products, energy carriers (such as biogas, biohydrogen, bioethanol, biochar, etc.) and value-added products (such as biomaterials, fertilizers, chemicals, and substances for use in the pharmaceutical industry), as building blocks for further processing [9]. Thus, biorefinery provides an opportunity to replace materials, chemicals, and fuels currently produced using fossil resources with renewable and sustainable biomass alternatives [10].

To make the bioeconomy truly sustainable, it is crucial to avoid mistakes of the existing linear economy model and develop new technologies and industries that consider the life cycle perspective and integrate them into the circular bioeconomy model [11]. The goal of circular bioeconomy is to enhance resource recovery by promoting the recycling of used materials and substances while minimising resource consumption and waste generation. The circular bioeconomy (CBE) thus applies the concepts to the use and processing of biomass aiming to produce energy, substances, and materials through biorefinery principles, at the same time limiting waste generation [12,13]. Therefore, the CBE concept encompasses all three segments of sustainable development by addressing economic, environmental, and social aspects [13].

Forestry can be considered a key resource for the bioeconomy as it provides a major source of timber, energy, chemicals (including fine chemicals), and various other products. Trees of industrial significance in forestry can be grouped roughly as coniferous and deciduous, with their differing properties and composition reflecting diverse marketing and utilization areas. These distinctions also highlight their significance as a source material for the bioeconomy. 

Worldwide, there are more than 600 coniferous tree species [14]. Besides their aesthetic value and provided ecosystem services, they are a major tree species extensively used in forestry, especially in northern European Union (EU) areas. The forested areas cover 160 million hectares, with coniferous tree species, such as Scots pine (*Pinus sylvestris*), Maritime pine (*Pinus pinaster*), Norway spruce (*Picea* spp., *Abies* spp.), and others [15] dominating the forests, particularly in Northern European regions. In 2021, coniferous trees accounted for 69% of all roundwood harvested in the forests and other wooded land of EU countries [16]. Forests cover more than half of the territory of Finland (66%), Sweden (63%), Estonia (54%), and Latvia (53%) [17]. In 2021, roundwood production in the EU reached an estimated 507 million m^3^ with significant contributions from Baltic Sea countries, such as Finland with 66.7, Sweden with 77, Latvia with 15.9, Lithuania with 6.6, Estonia with 10.6, and Poland with 43 million m^3^ of roundwood [16]. Timber production is expected to increase due to the Green Deal’s emphasis on replacing fossil fuels with renewable resources and the bioeconomy’s reliance on biomass resources from agriculture and forestry [18,19]. Conifers also serve as a significant source of non-timber products, including extracts, and resins which have many applications. Coniferous non-timber products include adhesives, food additives, products for cosmetics, and food. Conifers are the source of many substances with active applications in the pharmaceutical industry [20].

Conifer genomics [21] stresses the uniqueness of this plant group, considering its age dating from the Carboniferous period (c. 310 mya.) and its high number of individual contemporary species. Conifers have larger genomes compared to angiosperms with sizes ranging from 6.5 Gb in *Lepidothamnus intermedius* to 37 Gb in *Pinus ayacahuite* [22]. Conifers have several mechanisms providing a high adaptive response to climate change and pests. Their secondary metabolites contain a high number of biologically active substances that act as a chemical defence system. Conifers commonly produce exudates containing resins (consisting of abietane-, labdane-, and pimarane-type diterpenoids) as well as volatile substances that repel insects and other invaders, while also protecting the wood from further damage [23]. Conifer needles are covered with dense protective wax layers, which contain substances that deter consumption by predators. The highly efficient chemical defence system of conifers is believed to consist of thousands of substances, with resin acids, terpenes, and volatile substances playing key roles [24]. These substances in high concentrations can be found in bark and needles as well as in branches of trees. The direct-infusion ultrahigh-resolution Fourier-transform ion cyclotron resonance mass spectrometry (FT-ICR MS) has allowed the detection of approximately 3000 different compounds with diverse pharmaceutical and nutritional properties in the needles of northern conifers [25]. While there are significant differences in secondary metabolites among different conifer species, the overall group composition involved in the chemical defence system is relatively similar. An important group of conifer secondary metabolites are polyphenolics and tannins [26].

Besides the main product, timber, forestry produces significant amounts of various biomass side streams, often considered as forestry wastes in amounts up to 30–50% of the produced timber mass [27]. In another study, even higher proportions of the coniferous tree biomass, such as logging residues, accounted for as much as 78% of the merchantable roundwood produced [28]. Forestry side streams encompass cones, bark, and needles (small branches and needles collectively referred to as greenery), with greenery constituting the largest portion of the forestry side stream biomass. 

Coniferous greenery valorization should be conducted following a value cascading approach, adhering to the bio-based value pyramid (Figure 1) and the waste hierarchy, moving from high-value to lower-value biomass applications. However, considering the substantial amounts of forestry biomass side streams, applications on the lower part of the bio-based value pyramid could have a separate interest, considering both environmental and economic aspects. 

Coniferous tree greenery has been used for energy production, and gasification is considered a promising approach for generating bioenergy [30]. Another thermochemical processing approach, pyrolysis, has been proposed to produce biochar from coniferous needles, which can be used for energy production, in agriculture, and as a sorbent [31,32,33]. Coniferous needles have also been investigated for bioethanol production through enzymatic or chemical hydrolysis. However, the yield obtained cannot be considered as high (bioethanol yield of 3.89%, compared to wheat at 39% and corn at 41%) due to the low cellulose content, and other components of the needles appear to inhibit the fermentation process [34,35]. Moreover, coniferous needles have been used as fibres in combination with synthetic (urea-formaldehyde, resorcinol-formaldehyde, and others) and biodegradable (poly lactic acid and others) polymers or concrete to develop composites with enhanced mechanical properties [36,37]. The length and pre-processing of the needles play a crucial role in the mechanical properties of the obtained composite materials [38]. While biorefinery strategies for lignocellulosic biomass have been well elaborated, approaches for utilising coniferous biomass side streams, particularly needles, are not as extensively developed.

Considering the unique and highly complex chemical composition of coniferous needle/greenery biomass, a highly prospective biorefinery strategy can be pursued to target the top segments of the bio-based value pyramid. This approach involves coniferous biomass side streams as a highly prospective source of diverse chemical substances, which can potentially replace fossil material-based chemicals, serve as building blocks, be used in food and feed, and act as fine chemicals (Figure 1). This study aims to review the existing biorefinery methods for coniferous tree forestry biomass side streams (greenery/needles) and examine the potential to develop further biorefinery strategies for obtaining value-added products.

## 2. Chemical Diversity of Coniferous Needles and Wood Greenery

The studies of the chemical composition of coniferous needles/greenery started as early as the start of the 20th century, but comparable results were obtained after the Second World War. The classification of coniferous needles/greenery largely depends on the intended use, although a basic classification can be based on groups of components: alkanes, alkenes, aromatic substances, lipids, fats, waxes, carbohydrates, polysaccharides, lignins, polyphenolics, proteins, alkaloids, and more. However, from the perspective of coniferous needle/greenery biorefinery, the classification can be conducted based on the solubility of solvents that are prospective for biorefinery purposes: (1) substances soluble in low-polarity solvents (lipids), (2) substances soluble in water, and (3) substances insoluble in water and low-polarity solvents. 

The main components of the coniferous greenery/needles of pine (*Pinus sylvestris* L.), spruce (*Picea abies* L.), and Siberian fir (*Abies sibirica* L.) are cellulose and lignin. They also contain significant amounts of proteins, photosynthetic pigments, and carotenoids. However, through extraction, it is possible to obtain up to 35% of substances soluble in the selected solvent (Table 1.).

The CuO oxidation method revealed that the amount of polyphenols in the needle lignins was quite similar for spruce and pine, approximately 23 mg/g [40]. The main phenolic components identified were vanillin and *p*-coumaric acid, approximately 15 mg/g and 5 mg/g, respectively. Smaller amounts of Vanillic acid, ferulic acid, and *p*-hydroxybenzaldehyde were also detected. It was observed that Vanillic acid and *p*-hydroxybenzaldehyde were higher in spruce needles, measuring 2.9 mg/g and 1.4 mg/g, respectively, compared to 2.1 mg/g and 0.3 mg/g in pine needles.

### 2.1. Lipid-Soluble Components of Coniferous Needles/Greenery

The majority of components in coniferous greenery/needles that are soluble in low-polarity solvents (alkanes, lower aromatics, ethers, chlorinated solvents) belong to lipids. Lipids are considered both primary metabolites, as they are essential for plant growth and development, and secondary metabolites, as they serve functions that help the plants to better interact with their environment. The definition of lipids is generally broad, encompassing hydrophobic biological substances that are often soluble in organic solvents [41]. Numerous lipids contribute to cell metabolism regulation acting as signalling molecules. Lipids are the primary form of energy storage in plants and animals. The main form of energy storage in plants is glycerolipids. Due to the rapid development of analytical techniques, there are currently 45,648 unique lipid structures identified and classified [42]. In recent years, with the discovery of numerous novel lipid structures, more attention has been given to the classification of lipids, resulting in their division into eight categories—fatty acyls, glycerolipids, glycerophospholipids, sphingolipids, sterol lipids, prenol lipids, saccharolipids, and polyketides [43].

**Essential oils** are a heterogeneous group of substances soluble in low-polarity solvents, and they are characterized by their low boiling temperature or volatility with water vapour. Aliphatic hydrocarbons, such as myrcene (**1**), and aromatic hydrocarbons, such as *p*-cymene (**2**), are examples of compounds found in the essential oils [44].



A significant component of coniferous essential oils is **terpenes** [45,46]. Terpenes can be considered derivatives of isoprene (C_5_H_10_), and the most important classes are monoterpenes, sesquiterpenes (C_15_H_24_), and diterpenes (C_20_H_32_). Among them, limonene (**3**), terpinenes (*β*-, γ-, δ-, α-terpinene–(**4**)), and phellandrenes (*β*-, α-phellandrene–(**5**)) are found in high concentrations. Bicyclic terpenes, such as α- and *β*-pinenes (α-pinene–(**6**), 3-carene (**7**), and camphene (**8**)), are also significant components of coniferous essential oils.



Among the sesquiterpenes, some of the most important ones include cadinene (**9**), muurolane (**10**), amorphane (**11**), and others [20,47,48].



In addition to terpene derivatives, coniferous essential oils also contain oxygen-containing derivatives of terpenes, such as linalool (**12**), borneol (**13**), terpineols (**14**), α-terpineol, citral (**15**), and others [49]. Furthermore, aldehydes; phenolics; and, to a lesser extent, alcohols, acids, and other groups of substances can be found in smaller amounts within these oils [43].



**Carotenoids** (tetraterpenoids) are important pigments produced in relatively high concentrations in coniferous needles. Among the carotenoids, α-carotene (**16**) and β-carotene have been found in the highest concentrations [50]. Additionally, zeaxanthin (**17**) and lutein (**18**) have also been detected in coniferous needles [44].



**Green pigments** are vital for photosynthesis in conifers, with the most important pigment being chlorophyll-a. However, in addition to chlorophyll-a, other pigments, such as pheophytin and chlorophyllides, have also been found [50].

Coniferous needles also contain **lipid vitamins,** including Vitamins A, E, K, and D.

**Resin acids** are a group of substances common in conifers and play a significant role in their defence against parasites and infections [20,44]. Resin acids can be considered as oxygen derivatives of diterpenes and they contain partially reduced phenanthrene rings. The most abundant resin acids in conifers are abietic acid (**19**), pimaric acid (**20**), levopimaric acid (**21**), sandaracopimaric acid (**22**), and agathic acid (**23**).



**Alcohol derivatives** of terpenes are found in high concentrations in coniferous needles and are represented by labdane alcohols epimanool (**24**), cis-abienol (**25**), iso-abienol (**26**), phytosterols β-sitosterol (**27**), and methylcycloartenol (**28**).



**Polyprenols** are long-chain linear polymers consisting of several up to more than 100 isoprene units [44]. Two main types of polyisoprenoid alcohols have been described, so far differing in the hydrogenation status of their OH-terminal: (1) polyprenols (α-unsaturated), which contain the isoprene unit, and (2) dolichols (α-saturated). Polyprenols (**29**) are found in conifer photosynthetic tissues (needles) and bacterial cells, while dolichols (**30**) are typical animal and yeast lipids, and they have also been found in plant roots [51]. Polyprenols can be found as free alcohols and/or esters with carboxylic acids while a fraction (usually a small portion) is also found in the form of phosphates [51].



Among coniferous lipids, glycerolipids, glycerophospholipids, and sphingolipids can also be found.

**Waxes** play a significant protective function in coniferous needles. They cover the surface of the needles as epicuticular waxes and can also be found in cellular membranes [52]. Waxes consist of various components, including the highest hydrocarbons (C_21_–C_35_), primary and secondary alcohols (C_12_–C_32_), and diols (C_12_–C_31_, fatty acids, resin acids, ω-hydroxy acids as well as their esters and ethers) [53]. Among these components, nonacosan-10-ol has been identified in high concentrations, accounting for up to 60% of the wax mass [54,55,56]. Coniferous waxes have been suggested to create superhydrophobic coatings [57].

In addition to lipid-soluble substances, pine needles also contain large amounts of water-soluble substances.

### 2.2. Water-Soluble Components of Coniferous Needles/Greenery

Water, especially hot water, is a more powerful solvent than apolar solvents used for the extraction of lipids. With water, it is possible to extract up to 40% of the needle/greenery biomass. As a result, the obtained extracts contain a high number of different groups of substances representing both primary and secondary metabolites. Furthermore, water is an environmentally safe solvent. The obtained aqueous extracts can be used more extensively, as additional solvent removal is not required. On the other hand, water extraction may require high temperatures or a prolonged extraction time, which can lead to the extraction of unwanted metabolites such as sugars and salts. This can reduce the cost-effectiveness of the process.

**Vitamins** in coniferous needles are presented in high concentrations, and, amongst them, Vitamin C (ascorbic acid) is present in the highest concentrations of 200–250 mg/100 g of needles [43]. This concentration is comparable to sources like blackcurrants, lemon, and other food sources. Vitamin B_1_ (thiamine) can also be found in aqueous extracts, typically at concentrations of approximately 1–2 mg/100 g of needles, while Vitamin B_2_ (riboflavin) is found at levels of 0.5–1 mg/100 g. Other vitamins such as B_5_ (pantothenic acid), Vitamin PP (nicotinamide), B_9_ (folic acid) (4 mg/kg), Vitamin B_6_ (pyridoxine) (2–20 mg/kg), and Vitamin H (biotin) (0.5–1.5 mg/kg) have also been identified [58,59,60]. Therefore, the extraction of vitamins from coniferous biomass and the maximization of their value should not be overlooked. Presently, there is a lack of comprehensive scientific data in this area. However, it can be presumed that aqueous extracts enriched with vitamins can be utilized as ingredients for pharmaceutical, food, and nutraceutical products.

**Polyphenols** are another group of bioactive compounds found in coniferous biomass. They act as a preventative measure for cells against oxidative stress and free radical formation in biological systems. Due to their unique properties, polyphenols have a wide range of potential applications in pharmaceuticals, food, and nutraceutical products. They are well known for their antibacterial [61], antioxidant [58], anti-inflammatory, regulative, and neuroprotective effects. Structurally, these compounds can be divided into two broad classes: phenolic acids (primarily hydroxybenzoic acid and hydroxycinnamic acid) and flavonoids. Phenolic acids typically have a single benzene ring, while flavonoids have a basic structure consisting of 15 carbon atoms arranged in three rings (C_6_–C_3_–C_6_), labelled A, C, and B, respectively [62].

Coniferous needles’ aqueous extracts contain major groups of polyphenolics, which can exist in both free form and bound to carbohydrate residues [63,64,65]. Phenolic acids and their derivatives, such as hydroxycinnamic and hydroxybenzoic acids, have been identified in coniferous needle extracts, both in free form and bound as glycosides. Examples include quinic acid, cinnamic acid, ferulic acid, diferulic acid, coumaric acid, sinapinic acid, caffeic acid, and chlorogenic acid as well as 3-*p*-coumaroylquinic acid [48]. Additionally, various flavonoids have been found including kaempferol (in the highest concentrations), quercetin, isorhamnetin, myricetin, laricitrin, syringetin, and their derivatives. Among flavan-3-ols, catechin is the most abundant along with gallocatechol, naringenin, and apigenin both in free and glucoside forms. Proanthocyanidins formed from catechin or gallocatechin were also detected, including procyanidin A2, procyanidin B1, procyanidin trimer C1, prodelphinidin A, and prodelphinidin [48]. Stilbenes and tannins are common components of coniferous bark, but they can also be found in needles as free aglycones and the corresponding glucosides. The major stilbenes found in spruce needles, namely astrigin, piceid, and isorhapotin, were reported by Metsämuuronen and Sirén (2019) [66].

From lignans in needle aqueous extracts, taxiresinol, lariciresinol, pinoresinol, secoisolariciresinol, and hydroxymataresinol have been identified [59,61,67,68]. It has also been observed that the extraction of polyphenolic compounds from coniferous needles is affected by the preparation of the raw material and extraction conditions [58,68]. Comparing freeze-drying, vacuum-drying, and natural air-drying, freeze-drying yields the highest amount of polyphenol compounds. However, the total quantity of phenolic compounds varied from 0.89 to 1.53 g/kg dw, depending on whether natural air-drying or vacuum-drying was employed. This understanding enables the optimization of coniferous biomass preparation for polar extraction of polyphenolic compounds in a more cost-effective and environmentally friendly manner.

**Carbohydrates** in aqueous extracts of coniferous needles consist of mono-, di-, and polysaccharides. Monosaccharides including pentoses (arabinose, xylose, ribose) and hexoses (glucose, fructose, galactose, mannose, and others) were found [69,70]. Disaccharides such as sucrose, maltose, cellobiose, and rutinose have also been identified. The concentrations of water-soluble carbohydrates in coniferous needles vary with changing seasons [71,72]. Dry coniferous needles may contain approximately 28 g/kg glucose, 46 g/kg fructose, or 1.9 g/kg inositol, which the amounts varying depending on the season or the age of the tree [73]. In recent decades, researchers have focused on carbohydrates as forest soil organic matter [74]. Additionally, limited water availability can lead to increased isotope values (δ^13^C) in carbohydrates found in tree needles and may negatively impact tree growth [75,76].

**Other biologically active organic compounds**. Aqueous extracts of coniferous needles also contain other biologically active organic compounds that have the potential to be utilized in the production of value-added products. These compounds include sugar acids, such as quinic acid, threonic acid, gluconic acid, glucaric acid, xylonic acid, and 4-O-methyl-α-D-glucuronic acid, which are present in varying quantities. Quinic acid (cyclitol) (**31**) has been found in relatively high concentrations, and D-glucaric acid is abundant in the larch extracts [59]. Pinitol (**32**), one of several cyclitols specific to coniferous needles, is found at approximately 28 g/kg [73].

Additionally, shikimic acid (**33**), a significant metabolite for lignin synthesis, has been discovered in high concentrations [77].



In recent decades, there has been significant research interest in shikimic acid (3,4,5-trihydroxy-1-cyclohexene-1-carboxylic acid) due to its use as a base material for the synthesis of a neuraminidase inhibitor medicine. This medicine is utilized in the treatment and prevention of Influenza A and Influenza B [78]. The usage of water instead of organic solvent at relatively low temperatures has shown the ability to extract around 90% of the shikimic acid present in the biomass [79]. Studies have reported concentrations of shikimic acid extracted from coniferous needles ranging from 16 to 100 g/kg of biomass [77,78,80,81]. Therefore, researchers are actively seeking new sources for obtaining shikimic acid, and conifer needles have emerged as a potential source [77].

Coniferous needles contain a range of **alkaloids**, which are naturally occurring chemical compounds that typically contain basic nitrogen atoms. These alkaloids have the potential to prevent the onset of various degenerative diseases by scavenging free radicals or binding with the oxidative reaction catalysts. Aqueous extracts of coniferous needles have been found to contain piperidine alkaloids, specifically α-pipecoline (**34**), cis-pinidine (**35**), and others [59,82,83].
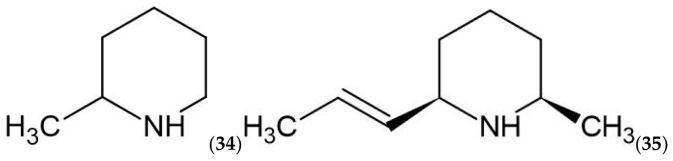


The forestry sector plays a significant role in the economies and ecologies of many countries [16]. As illustrated in the preceding discussion’s part, the residues and by-products of coniferous needles are rich in a wide range of industrially relevant compounds, highlighting their potential as valuable resources within the circular bioeconomy. Given the high chemical diversity of coniferous needles/greenery, their potential for use in the circular bioeconomy is significant. However, to fully exploit this potential, further development of biorefinery approaches is necessary to effectively valorize the side streams of forestry biomass.

## 3. Biorefinery Approaches for the Extraction of Coniferous Needles and Wood Greenery

The biorefinery approaches for forestry greenery/needles can be classified accordingly to the initial stage of the needle/greenery processing, with the extraction process (the type of solvent) playing a key role in determining the biorefinery strategy. Essential oils, which are a special group of components found in coniferous needles, are typically isolated based on the volatility of these substances.

The biorefinery approaches employed for coniferous needles/greenery are highly dependent on the specific substrate used, whether it is needles or a mixture of needles with small branches. The efficiency of the biorefinery is also influenced by factors such as the intensity of biomass disintegration, drying methods, and the use of wet biomass. Several studies have examined the changes in the chemical composition of coniferous needle/greenery biomass, including the analysis of seasonal variation in needle composition and the effect of tree age [84,85]. The concentrations of carbohydrates and amino acids are found to be the most sensitive to seasonal changes, while lipids, fatty acids, and resin acids exhibit greater resistance to such variations. Nonetheless, seasonal effects and the age of the trees still have relatively significant impacts on the composition of coniferous needle/greenery biomass.

### 3.1. Coniferous Needle Essential Oils

Coniferous needles contain significant amounts of essential oils. A comprehensive review of pine needle essential oils, including their composition, extraction methods, and areas of application, is provided by Silori et al. (2019) [86]. The concentration of essential oils in coniferous needles depends on various factors such as the tree species [44], geographical location, and climatic factors [87]. Additionally, tree age has been found to influence the composition of oils in conifers, as demonstrated by the increasing concentration of essential oils with age, as observed in *Pinus sylvestris* [88]. Essential oils from coniferous needles can be obtained through extraction of the essential oils using low-polarity solvents such as hexane, petrol ether, or supercritical CO_2_. Another approach is hydrodistillation [89,90]. In this method, the fresh needle samples are cut into small pieces and subjected to hydrodistillation. The isolated essential oils are then dissolved in pentane and dried using anhydrous sodium sulphate [91].

Needle essential oils have been recognized for their strong therapeutic properties [92]. The research has demonstrated the beneficial effects of pine oil in relieving bronchial infections [93]. Pine oil has been proven to be a potent agent against harmful organisms and can aid the body in fighting infections. In addition to their therapeutic applications, pine oils are widely used as fragrances in the cosmetic industry, as flavouring additives in the food and beverage industry, and as scenting agents in various household products. Furthermore, pine oils serve as intermediates in the synthesis of perfume chemicals [94]. For example, limonene is used to commercially synthesize carvone [86]. Moreover, pine oils are used for medicinal purposes in aromatherapy, where they act as carminative (relieving flatulence), rubefacient (stimulating blood circulation), emmenagogue (promoting menstruation), and abortifacient (inducing abortion) agents. Traditional remedies utilizing various pine preparations have been employed for the treatment of different ailments, such as ptilosis, dermatitis, toothache, etc. [95]. Numerous studies investigating the pharmacological activities of pine species have shown that turpentine, their extracts, and isolated compounds exhibit antioxidant, antiviral, analgesic, anti-inflammatory, cytotoxic, and/or antimicrobial properties [96]. Furthermore, pine essential oils have exhibited significant biological activities, such as antifungal, acaricidal (against mites), and antiplatelet effects. Due to their pleasant fragrance, they are commonly used in cosmetic industries. Conifer essential oils have also been proven to hold an important role in the defence system of conifers against numerous herbivorous insects and pathogens [97].

The essential oil content in *Pinus sylvestris* varies between 0.22 and 0.82%. The highest volatile levels of oil are typically observed during the summer season [98]. The primary constituents in *P. sylvestris* needle oils are α-pinene, camphene, and β-pinene. The ratios of α -pinene, β -pinene, and camphene in essential oil samples range from 19.44 to 56.88%, 2.87 to 7.09%, and 0.44 to 16.84%, respectively. In *Pinus canariensis*, steam distillation of needles revealed the presence of 75 compounds. The majority of the compounds were monoterpenes (43%), followed by sesquiterpenes, diterpenes, alkane derivatives, and benzyl esters of benzoic and salicylic acid [99]. According to Silori et al. [86], the major components in pine essential oils include β-pinene, camphene, α-pinene, sabinene, 3-carene, myrcene, α-terpineol, terpinolene, limonene, bornyl acetate, caryophyllene, terpinene-4-ol, γ-muurolene, phellandrene, α-terpinene, thujene, γ-terpinene, p-cymene, germacrene D, and spathulenol.

### 3.2. Preparation and Properties of Aqueous Extracts

Aqueous extraction has been used for the processing of needle/greenery biomass as one of the first methods to obtain valuable components from needles. Historically, green needles were treated with hot water to obtain teas rich in vitamins (initially focusing on Vitamin C) to combat Vitamin C deficiency during the winter season or during long sea voyages [100]. This practice also helped in the prevention of scurvy [101].

Water, especially hot water, can be considered one of the most powerful and sustainable solvents for the processing of coniferous biomass. It provides the possibility to isolate mineral substances, amino acids, proteins, monosaccharides, low-molecular-weight polysaccharides, vitamins, a significant part of polyphenolics, low-molecular-weight organic acids, and many other substances [102]. Technologies, such as ultrasound, cavitation, treatment with microwaves, and the use of pressurized reactors operating under subcritical water conditions, have significantly improved the efficiency of aqueous extraction. Nowadays, these technologies can be applied on an industrial scale [103,104]. Due to the potency of aqueous extraction and the presence of water-soluble substances in the needle biomass, it is possible to obtain the highest concentrations of extractives per mass of needles, reaching up to 35%.

Aqueous extracts are complex mixtures, containing various groups of substances, which, in accordance with the biorefinery principles, necessitate subsequent fractionation. These extracts are utilized either in their entirety or after evaporation, without further fractionation, due to their demonstrated antimicrobial, antiviral, radical scavenging, and other activities [81]. The employed aqueous extraction procedures are straightforward, easily scalable, and prospective for industrial applications. For instance, the water-soluble extract from pine needles of *Cedrus deodara* demonstrates a high antibacterial activity against food-borne bacteria including *Escherichia coli*, *Proteus vulgaris*, *Staphylococcus aureus*, *Bacillus subtilis,* and others. Shikimic acid has been identified as the main antibacterial compound, thus endorsing the use of aqueous extracts as a novel source of natural antibacterial agents applicable in the food industries [105]. For isolation of shikimic acid, needles from *Pinus elliottii* are ground and extracted with hot water, and the extractions are repeated several times. Employing intensive extraction methods allows for a significant increase in extraction yields [106]. Following evaporation, the extracts are treated with activated carbon to obtain a purified material [107], although the use of ion exchange resins is more efficient. Anion-exchange resins selectively absorb shikimic and other organic acids, which are subsequently desorbed using an alkaline eluent, followed by evaporation and crystallization [79].

Aqueous extracts have been shown to possess antibrowning and antimicrobial activities, making them suitable for food preservation [105]. The efficiency of the extraction process can be enhanced through process intensification and improving the properties of the obtained products. An intensive extraction process based on controlled hydrodynamic cavitation provided an opportunity to obtain aqueous extracts from silver fir (*Abies alba* Mill.) with enhanced antioxidant activity [108]. Ongoing research aims to explore new applications for the obtained extracts, as they have demonstrated an effective inhibitory effect on seed germination (potentially attributed to the presence of terpenes), indicating their potential for weed control [109]. Polysaccharides have been isolated and characterized from pine needles of *Cedrus deodara* using aqueous extraction. The isolated acidic heteropolysaccharide consists of glucose, arabinose, mannose, and xylose with a molecular weight of 1.53 × 10^4^ Da, demonstrating high antioxidant activity, the ability to scavenge free radicals, and the potential to inhibit oxidative damage to DNA and cells. It is suggested that this polysaccharide could serve as a potential antioxidant agent for developing functional foods [105]. Furthermore, aqueous extracts obtained through cavitation have been successfully incorporated into bread, resulting in value-added and functional bread products [110]. Aqueous extracts obtained both from fresh and dried pine needles *(Pinus densiflora*) exhibit significant reductive power, inhibition of lipid peroxidation, and the ability to scavenge free radicals, comparable to those of known antioxidants—phenolic compounds. Thus, these extracts can be considered potent in vitro antioxidants, both in non-cellular and cellular systems [111]. The antioxidant activity of these extracts plays a crucial role in their potential applications, particularly in the food industry [112]. One notable application in the food industry involves the preparation of pine needle extract using ultrasound treatment. This extract is mixed with beeswax and used as a coating to preserve cheese, imparting antioxidant and antibacterial properties while inhibiting the growth of moulds and extending the shelf life of the cheese [113]. Aqueous extracts of needles from *Pinus massoniana* Lamb. contain several organic acids, with acetic acid being the dominant component at a concentration of 25%. Hexadecanoic acid is another significant acid present in the extracts, constituting 19% of the composition. Additionally, other organic acids such as citric acid, succinic acid, malonic acid, malic acid, oxalic acid, and others [81] are present in lower concentrations. These organic acids contribute to the antibacterial activity exhibited by these extracts [114]. Furthermore, a separate study highlighted the high complexity of aqueous extracts, revealing the presence of several aliphatic mono- and dicarboxylic acids, hydroxycarboxylic acids, aromatic and alkyl-aromatic acids, and hydroxyl derivatives of benzaldehyde and acetophenone [115].

### 3.3. Alcoholic Extraction

Alcoholic extraction is commonly employed for extracting coniferous needles, utilizing the lowest alcohols such as methanol, ethanol, n-propanol, and isopropanol. These alcohols are preferred due to their low toxicity (except for methanol), cost-effectiveness, and safety considerations, while providing extraction yields comparable to those achieved with hot water extraction (around 30–40%). The lower alcohols exhibit excellent permeating ability through the cell walls, allowing for the dissolution of medium- and low-polarity substances from the needles. Consequently, they can effectively extract carbohydrates, polyphenolics, many resin acids, photosynthetic pigments, amino acids, and many other groups of substances [116]. However, the high extraction efficiency of alcoholic solvents presents a drawback, as it transfers a multitude of substance groups from the needle biomass to the extract, resulting in a complex and multi-step fractionation process. In comparison to aqueous extraction, alcoholic extraction also extracts substances with lower polarity such as terpenes. However, highly polar substances in the extract are transferred to a lesser extent.

Efforts have been made to directly utilize alcoholic extracts of coniferous needles/greenery, due to the significant transfer of individual substances in the alcohol, demonstrating the high potential for application of such extracts. For instance, methanol or ethanol of Chir pine (*Pinus roxburghii*) needles was obtained, containing alcohols, ketones, terpenes, and polyphenols. These extracts demonstrated herbicidal activity against two weed species as well as insecticidal activity, indicating their prospective use as bio-pesticides for weed and insect pest management [117]. In addition, alcoholic extracts of *Pinus roxburghii* needle were investigated for antidyslipidemic activity in the context of a high-fat diet. The extracts displayed antioxidant activity and exhibited an impact on lipid metabolism, suggesting their potential in managing lipidaemia [118].

In the processing of coniferous needles, it is common to utilize aqueous alcoholic (methanolic, ethanolic) mixtures. By varying the water-to-alcohol concentrations, a certain degree of selectivity of different groups of needle components can be achieved. The treatment with ultrasound has shown promise in increasing extraction yields [119]. For instance, an aqueous methanolic extract of *Pinus densiflora* needles was partitioned with hexane, ethylacetate, and n-butanol. The resulting fraction exhibited anti-inflammatory effects with respect to skin inflammation [120]. Furthermore, aqueous methanolic (with 70% methanol) extracts from pine needles of *Cedrus deodara*, rich in phenolic compounds and possessing high antioxidant capacity, have been used in the preparation of chitosan-based antioxidant films for food packaging purposes [121].

Isopropanol has emerged as a prospective extractant for the processing of coniferous needles. It possesses several advantageous properties, including the ability to extract lower-polarity substances, cost-effectiveness, favourable recycling possibilities, and high overall extraction capacity [122]. Using iso-propanol as the extractant, extracts can be obtained from needles/greenery with yields up to 35% of the dry mass [84]. Considering the high number of substances that can be extracted with isopropanol, subsequent fractionation steps are necessary to obtain groups of substances with high application potential for various industries. After the evaporation step, which allows for the regeneration of iso-propanol, the residue is re-extracted with petroleum ether. This process aims to obtain substances with low polarity, which can then be treated with an acid and subsequently neutralized. Through this procedure, the extracted compounds can be further divided into groups of acids and neutral substances for targeted applications. In the fraction of neutral substances obtained from the iso-propanol extraction of coniferous needles, various compounds are present, including sesquiterpenes, diterpenes, triterpenes, and polyprenols. To isolate the fraction of phenolic acids, the extract can be treated with NaHCO_3_, followed by acidification and transfer of free phenolic acids into diethyl ether. Alternatively, another approach involves re-extraction of the iso-propanol extracts with hydrocarbons such as gasoline and petrol ether, followed by hydrolysis with dilute sodium hydroxide. In this method, sodium salts of organic acids are transferred to the aqueous phase, while neutral substances remain in the organic solvent. The salts of organic acids should be converted into their acid form through treatment with 35% sulfuric acid before isolation. Furthermore, the neutral substances can be saponified using 35% NaOH, and after acidification, chlorophyllin can be isolated (Figure 2). Chromatographic fractionation can be employed to isolate a high number of terpenes, as well as individual resin acids, from the extracts [84]. A more simplified approach for processing iso-propanol extracts was suggested by Troshina and Roschin (2014). This method involves a two-stage refinery process, allowing for the extraction of free resin and fatty acids, with the identification and quantification of the neutral and main substances present [38].

Carbohydrate composition analysis of ethanol extracts from *Pinus nigra* and *Abies alba* using ^13^C NMR revealed that carbohydrates accounted for 15% to 35% of the crude extracts. Pinitol was found as the predominant constituent, accompanied by the presence of other carbohydrates such as arabinitol, mannitol, glucose, and fructose [70].

### 3.4. Lipid Extraction and Fractionation

From an application perspective, lipids constitute a significant group of components found in coniferous needles/greenery. To extract this group of substances from needles, the use of low-polarity solvents becomes necessary. Considering the substantial volumes of needles to be processed, hydrocarbons are preferred as extractants. Various solvents, including linear alkanes (hexane, octane), cycloalkanes (cyclohexane), chlorinated solvents and similar compounds have been studied for their ability to extract lipids and other low-polarity components from coniferous needle/greenery biomass. The yields of extractives using low-polarity solvents are generally significantly lower compared to the use of alcohols, alcohol/water mixtures, or water as extractants. Typically, the yield is <10% of the dry mass. When using low-polarity solvents, low-polarity substances like alkanes, fatty acids and their esters, diterpenes, sesquiterpenes, esters of resin acids, polyprenols, and others are transferred to the liquid phase. Furthermore, due to the co-extraction of multiple substance groups when using hydrocarbons, advanced biorefinery methods have been proposed and described in various studies [123,124,125] (see Figure 3).

The biorefinery of lipophilic extracts from coniferous greenery follows a conceptual process that includes several steps: (1) extraction with hydrocarbons; (2) separation of obtained extracts from needle mass; (3) isolation of needle waxes; (4) distillation with water vapour or solvent distillation to obtain essential oils; (5) extraction with subsequent saponification to produce so-called chlorophyll-carotene paste; (6) treatment of the extract from step 2 with a solution of alkalis (NaOH) to transfer organic acids in sodium form (solubilize) with following acidification and splitting into two fractions (balm paste and sodium chlorophyllin); and (7) re-extraction of the solution after step 2 with petrol to obtain a complex of provitamins. This detailed biorefinery strategy allows for the production of six products: essential oils, chlorophyll-carotene paste, balm paste, sodium chlorophyllin, and a complex of polyvitamins (Figure 3). This strategy was developed and implemented in the Soviet Union and products nowadays are produced in Russia and partly in Latvia, but it is not widely known in other countries.

However, there are two main problems associated with this biorefinery strategy:(1)Poor analytical characterization and identification of the obtained products and their individual substances: There is a lack of comprehensive analysis and identification of the specific compounds present in each fraction obtained from the biorefinery process. This limits the knowledge and understanding of the potential application areas and benefits of these products. Currently, only polyprenols have found a place in the market as food supplements with a potential application in the pharmaceutical industry;(2)Environmental and safety concerns related to the use of hydrocarbons: The extraction process involves the use of large amounts of hydrocarbon solvents at the first and other stages of extraction. This raises concerns regarding the environmental impact and working safety requirements. Furthermore, residues of hydrocarbons may remain in the final products, significantly hampering their application potential.

### 3.5. Emulsion Extraction

One of the major challenges in the development of biorefinery processes for coniferous greenery is the initial extraction stage. Traditional methods use hydrocarbon solvents such as industrial-scale gasoline to obtain low-polarity substances. However, this approach has drawbacks due to the large volumes of needle mass requiring significant amounts of solvents. Residues left after extraction still contain hazardous flammable solvent residues posing environmental and safety concerns. An alternative approach to address these challenges is the use of emulsion extraction [126]. In this method, coniferous greenery is treated with alkaline solutions such as NaOH 5% [127,128], employing intensive extraction techniques like cavitation or ultrasound treatment. During this process, natural surfactants in the greenery, such as salts of fatty and resin acids, are formed. These surfactants enable the formation of micelles, facilitating the transfer of lipids into the emulsion phase. Additionally, in an alkaline environment, complex esters undergo hydrolysis, contributing to the development of the emulsion by high-molecular alcohols present in the greenery [129]. The extraction process aims to obtain low-polarity substances, including polyprenols. Following the removal of residual greenery through filtration, the emulsion (aqueous phase) undergoes a second extraction with a low-polarity solvent, such as petroleum ether, albeit in significantly lower amounts compared to the direct extraction of greenery [130]. The lipid fraction obtained can be further processed using preparative chromatography to isolate polyprenols as the final product. The aqueous phase, upon acidification, can be re-extracted with diethyl ether to obtain acids, although their composition has not been much studied, and subsequently discarded [126]. Greenery residues after extraction are discarded as well. The efficiency of this emulsion extraction method relies on the intensity of the pretreatment of coniferous greenery (e.g., size of biomass particles) and the treatment process itself. Emulsion extraction allows for obtaining neutral substances (0.9–1.9% of the greenery dry mass), organic acids (2.7–3.2%), and polyprenols (0.13–0.23%) [126]. However, a significant portion of the extracted substances (up to 30% of the greenery mass) remains in the aqueous phase, and because their potential applications have not been studied, they are discarded. The acid fraction of the extraction contains diterpenes (isopimaric, abietic, dehydroabietic acids), as well as fatty acids. Additionally, *p*-hydroxybenzoic acid, ferulic acid, *p*-coumaric acids, and other compounds have been identified and quantified. The neutral fraction consists of diterpene alcohols (epimanool, dehydroabietinol, *p*-cymol) and polyprenols [131,132,133].

### 3.6. Extraction with Supercritical Fluids

As an environmentally friendly (non-toxic, non-flammable, physiologically compatible) extraction approach with potential uses in industry, extraction with supercritical CO_2_ (scCO_2_) is highly prospective as it has been widely demonstrated on different materials [134,135,136]. Unlike solvent extractions, where the solvent needs to be retrieved and re-used for further extractions, scCO_2_ extraction re-circulates the CO_2_ through the system without generating it during the extraction process. Moreover, the CO_2_ evaporates from the extract, leaving no harmful residues in the extracts, while solvents can leave behind residues introduced during the manufacturing process or added as solvent stabilizers. The advantages of scCO_2_ extraction, coupled with the development and increased availability of industrial-scale extraction units, make it a viable option for extracting lipid substances from diverse materials. This extraction method offers efficiency, a low carbon footprint, and reduced product costs. The obtained extracts exhibit a similar composition to those extracted using non-polar solvents (chloroform, hexane, petroleum ether, ethyl acetate), resulting in comparable extraction yields. Furthermore, the addition of co-solvents can enhance the extraction efficiency of polar compounds, for example, polyphenols.

Extraction with scCO_2_ has been widely applied to extract lipids and other low-polarity substances from coniferous tree greenery, particularly needles. Successfully, scCO_2_ has been used for the extraction of essential oils from conifers. For example, the yield of essential oils from oil from *Juniperus communis* needles was at a rate of 6.55 wt% relative to the initial mass, and even substances with low thermal stability were recovered [137]. In the case of *Pinus massoniana* needle extracts, supercritical CO_2_ extraction combined with an ethanol-methanol co-solvent yielded extracts with high antioxidant activity [138]. *Pinus nigra* yielded diterpenes with a yield of 1.60%, which have potential applications in perfumery and pharmaceutical industries [70]. A comparative study investigating the scCO_2_ potential and Soxhlet extraction [55] possibilities from various parts of Norway spruce (*Picea abies*) including needles, cones, bark, and branches indicated that the needle extract composition was represented mainly by terpenes (about 13.5 g/kg), sterols (9.2 g/kg), aromatic compounds (8.9 g/kg), nonacosan-10-ol (4.7 g/kg), fatty acids (oleic acid, n-hexadecanoic acid, and tetradecanoic acid), and resin acids (mainly dehydroabietic acid, about 0.9 and 3.8 g/kg, respectively).

Several studies focused on the extraction of polyprenols, compounds with high antiviral activity and potential diverse applications in medicine, from coniferous needles. A comparison between scO_2_ and solvent extraction from four different conifers revealed that *Pinus sylvestris* needles provided the highest polyprenol yield of 14.00 ± 0.4 mg/g dry weight when extracted with a hexane: acetone solvent. However, using SFE-CO_2_ extraction (at 200 bars, 70 °C, 7 h) with absolute ethanol as a cosolvent, a polyprenol yield of 6.35 ± 0.4 mg/g dry weight was obtained [139]. Importantly, the polyprenols obtained after single-stage saponification had acceptable purity. In another study [140], scCO_2_ extraction from spruce needles followed by saponification yielded polyprenol at a yield of 13.9 ± 0.6 mg/g dry weight. The extracts obtained were highly stable and free from organic solvents, making them suitable for direct use as food supplements or as a starting material for the preparation of semisynthetic polyisoprenoid derivatives, such as polyisoprenoid phosphates.

Another group of substances which can be obtained using scCO_2_ extraction are waxes. In the case of spruce species such as Norwegian spruce and Sitka spruce, the extraction of wax resulted in the isolation of nonacosan-10-ol, which is a hydrophobic alcohol with potential applications in superhydrophobic coatings. The yield of nonacosan-10-ol from the wax extracts was reported as 8070 μg/g of needles. To purify nonacosan-10-ol from the crude wax extracts, a simple rapid green re-crystallization method was employed, achieving a yield of approximately 44% from the crude wax [54].

Extraction using scCO_2_ is known for its ability to produce high-purity extracts without the need for extensive purification steps or the use of hydrocarbon or chlorinated solvents. However, one challenge that needs to be addressed is the utilization of coniferous biomass residues after extraction. So far, only a few studies have paid attention to this issue, suggesting the conversion of extraction residues into biochar and agricultural products [30,141].

## 4. Further Development of Coniferous Needle and Wood Greenery Biorefinery

Coniferous needles and greenery are highly prospective resources for further development of the bioeconomy, considering their volumes and the presence of many components with high application potential in different branches of the economy. The main factors affecting the potential of needle/greenery utilization are the large volumes of biomass and the lack of large-scale processing and utilization approaches that have been implemented so far. Because coniferous needles naturally decay slowly, their removal is not expected to cause harm or adverse impacts on forest ecosystems. Additionally, needles contain relatively low amounts of mineral substances, so their removal will not significantly affect the mineral matter balance in forest ecosystems. Furthermore, it is important to note that not every single needle can be removed from the forest floor, minimizing the impacts on growing forests. Any loss of mineral matter from the forests due to needle removal can be compensated through forest fertilization using biomass ashes, which are produced as a result of timber usage for energy production (Figure 4).

Still, there are remaining obstacles from the perspective of biomass utilization. One of the key challenges is the need to change existing forestry practices, ensuring possibilities to collect needle/greenery biomass. Currently, forestry activities primarily focus on timber production, while other biomass types are often considered waste. This poses a problem in obtaining “pure” needle biomass as activities such as the removal of larger branches can mix different biomass types together. Therefore, a major shift in the approach to tree biomass is urgently required. To drive progress in this regard, the implementation of legislative and economic instruments can play a crucial role in motivating forestry operators to adopt new practices and technologies for processing forest biomass.

The next issue related to the development of needle/greenery utilization as a biomass source for biorefinery is related to the pre-processing of the needles. Key steps in this process may involve drying or disintegration (milling, etc.). However, it is worth noting that there are existing facilities and technologies used in agriculture that can be adapted for the pre-processing of needle biomass. The key is to explore their application specificity for needle/greenery biomass, ensuring their suitability and effectiveness in this context.

The paradigms of plant biomass processing during the last decades have undergone major changes with respect to the need to consider environmental aspects and follow green chemistry principles, proposing “green extraction”. A definition has been suggested: “Green Extraction is based on the discovery and design of extraction processes which will reduce energy consumption, allow the use of alternative solvents and renewable natural products, and ensure a safe and high-quality extract/product” [142]. When designing extraction and purification processes of extracts from greenery/needles, it should be considered to exclude or reduce the use of toxic and hazardous solvents, such as organochlorinated solvents, benzene, and diethyl ether [143] (Figure 4). The replacement of dangerous solvents should affect not only the extraction stage but should also be considered throughout the full biorefinery process, supporting the development of a new paradigm in the processing of coniferous biomass. Green solvents, which can be produced using forest or agricultural biomass, include ethanol, methyl esters of fatty acids, glycerol, and others [142]. Further development of extraction techniques can provide new possibilities. Among these techniques, the widely used extraction with supercritical CO_2_, pressurized hot water extraction, subcritical water extraction, and solvent-free extraction techniques have demonstrated the benefits of their application [144,145]. For the extraction of natural products, green solvents, such as ionic liquids (ILs) [146], deep eutectic solvents (DESs), and natural deep eutectic solvents (NADESs), are becoming popular [147]. The known and recently suggested approaches for biomass processing demonstrate promising prospects for the processing of forestry as well other industry biomass side streams [148]. These approaches have the potential to enable large-scale production and yield a wide range of substances for various applications.

## Figures and Tables

**Figure 1 molecules-28-07085-f001:**
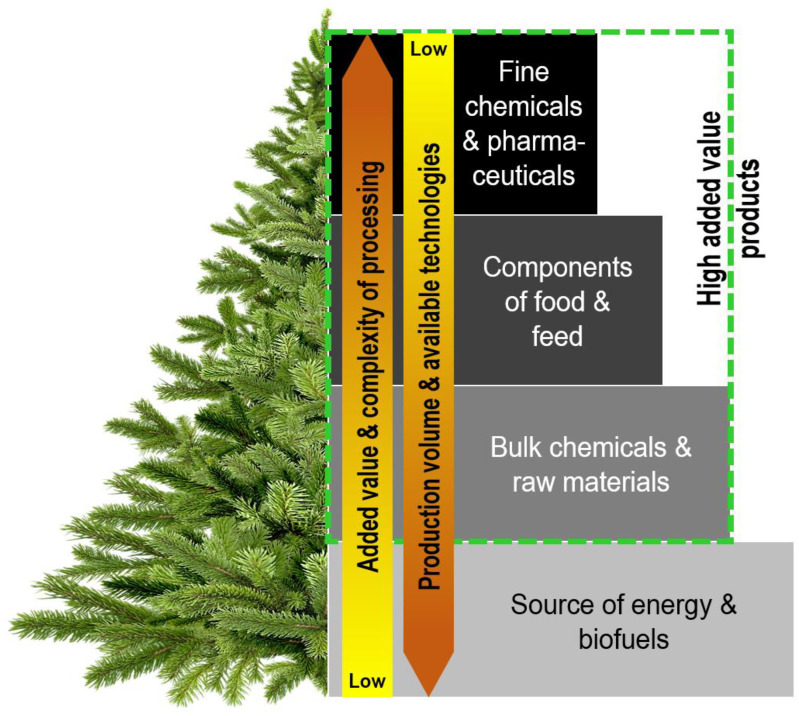
Bio-based value pyramid of forestry biomass side streams in the context of the scope of this study [13,29].

**Figure 2 molecules-28-07085-f002:**
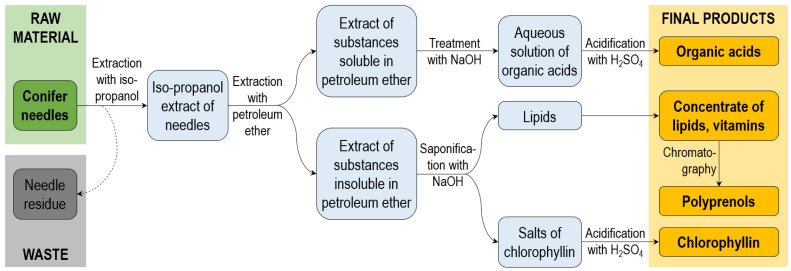
Biorefinery of isopropanol extract of coniferous needles [38,84].

**Figure 3 molecules-28-07085-f003:**
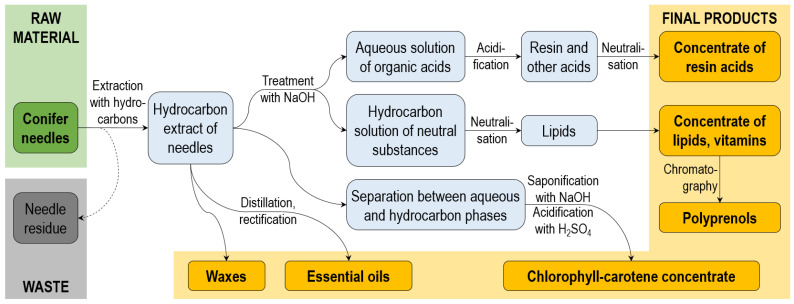
Biorefinery of isopropanol extract of coniferous needles [37].

**Figure 4 molecules-28-07085-f004:**
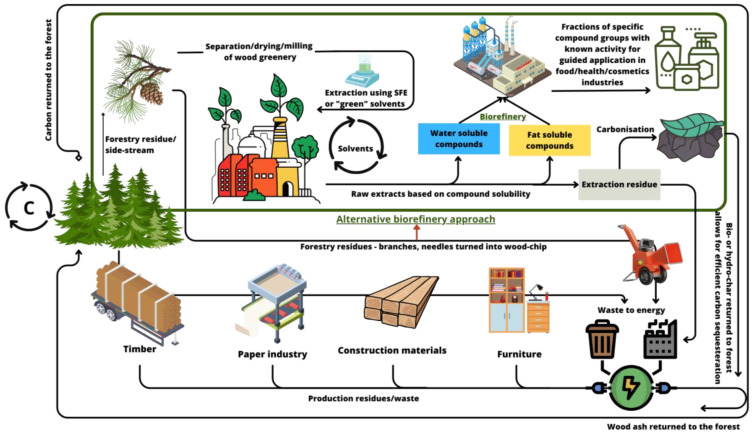
Proposed biorefinery approach for coniferous tree wood greenery to achieve carbon neutrality.

**Table 1 molecules-28-07085-t001:** Major components of coniferous greenery [38,39].

Components	Pine		Spruce		Siberian Fir	
	Needles	Greenery	Needles	Greenery	Needles	Greenery
**Cellulose, %**	28	20	19	25	22	24
**Lignin, %**	21	32	31	37	32	44
**Protein, %**	7.5	3	9	9	7	3
**Mineral substances, %**	2.5	2	2	2	4	3.5
**Chlorophyll, mg/kg**	5150	600	7580	1100	8978	1988
**Carotenoids, mg/kg**	168	16	179	10	236	12
**Substances, extracted with corresponding solvent, %**						
**Petrol ether**	15	9	7.5	4	10	13
**Chloroform**	9	13	13	5	19	17
**Ethanol**	34	26	37	16	39	23
**Isopropanol**	31	29	39	14	37	24
**Water (hot)**	38	26	-	-	35	11

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
