# Peer review of "Strategy of Coniferous Needle Biorefinery into Value-Added Products to Implement Circular Bioeconomy Concepts in Forestry Side Stream Utilization"

_molecules, 2023, doi:10.3390/molecules28207085_

Round 1

Reviewer 1 Report

-Page 1 Line 1-3  Title title. should it be added value product or value-added product. please recheck with other publication

-Page 2 Line 60-70  industry, as building blocks for further  processing and platform chemicals?, please check out the examples from these papers http://jase.tku.edu.tw/articles/jase-202402-27-2-0001, https://www.ncbi.nlm.nih.gov/pmc/articles/PMC6178844/, http://ojs.kmutnb.ac.th/index.php/ijst/article/view/3788/pdf_269,

-Page 2 Line 80 Worldwide, there are more than 600 coniferous tree species if possible , authors may add the quantitative data of coniferous biomass annually

-Page 3 Line 93-97   Conifers also serve as a significant source of non-timber products, including extracts please add reference for these sentences

-Page 3 Line 104-106 Conifers commonly produce exudates containing resins it will be good info if authors provide example of chemical compounds in exudate”

-Page 4 Line 139-140  However, the yield obtained cannot be considered as high please add more detail about this and compare this info with other plant

-Page 11 Line 364-365  The forestry sector plays a significant role in the economies and ecologies please give example

-Page 12 Line 408 pine oils serve as intermediates in the synthesis of perfume chemicals please give examples

-Page 13 Line 456-457 applications. For instance, the water-soluble extract from pine needles is there any report for antifungal?

-Page 14 Line 480 of 1.53 × 104 Da please recheck this

-Page 17 Line 597-598 This detailed biorefinery strategy allows for the production of six products: please add reference

-Page 18 Line 647  Extraction with supercritical fluids authors should add another table to summarize or compare each extraction solvents

minor revision is suggested

Author Response

REVIEWER 1
Dear
Reviewer 1,
On behalf of the authors of this review I would
l ike to thank you for the comments that you have
expressed. I believe that these implementations will improve the overall quality of the manuscript.
Please see the answers to the comments below
The title has been modified according to the suggestion.
By this the authors mean that there are different types of biomass that could contain substances that
could be used as platfrom chemicals chemicals that can serve as a substrate for the production of
various other higher value added products. Within our study we would like to demonstrate to use
forestry biomass side streams not only to obtain platform chemicals to preplace fossil reseource
chamicals, but rather to obtain high added value chemicals.
Thank you for the suggestion, however, such data is not available, especially in the terms of side
streams. Most of the forestry side streams are beeing used as fuel for co generation. This is even more
accurate for coniferous biomass that contains a lot of conifer needles. FAOStat only provides
information on forestry products such as veneer, pulpwood, sawlogs, and roundwood.
The reference has been added and the numbering of literature sources has been revised..
The importance of forestry sector has been discussed in the introduction part highlighting the importance of this industry within the region of Northern Europe. A reference has been added to this sentence. --Page 12 Line 408 pine oils serve as intermediates in the synthesis of perfume chemicals ““please give examples”” An example has been added with the corresponding reference. --Page 13 Line 456-457 applications. For instance, the water-soluble extract from pine needles ““is there any report for antifungal?”” While essential oils have reported antifungal activities, aqueous extracts are not reported to have high antifungal activities. --Page 14 Line 480 of 1.53 × 104 Da ““please recheck this”” Superscript has been added to 104 --Page 17 Line 597-598 This detailed biorefinery strategy allows for the production of six products: ““please add reference”” Reference has been added. --Page 18 Line 647 Extraction with supercritical fluids ““authors should add another table to summarize or compare each extraction solvents”” Indeed, this would be a useful metric of comparing the extraction methods side-by-side, however, due to the differences in the sample preparation (drying, extraction from wet biomass, extraction temperature, intensities, species in question) this could give a wrong ide to the reader about the most effective extraction techniques and methods. Also, considerations about the used solvents (petrochemicals or environmentally friendly solvents) could provide misleading information. Nevertheless, our research group is currently working on an article where several extraction methods will be tested using the same pine and spruce materials to extract resinous compounds, thus showing the most appropriate extraction methods and solvents used, based on their impact on the environment.
Yours sincerely,
Linards Klavins and the authors

Reviewer 2 Report

Slight modification needed

Author Response

REVIEWER 2

Dear Reviewer 1,

On behalf of the authors of this review I would like to thank you for the comments that you have expressed. I believe that these implementations will improve the overall quality of the manuscript.

Please see the answers to the comments below -

 The following are my comments on the manuscript:

  1. Title: Consider making it crisp.

The title has been revised according to reviewers comments.

  1. Abstract: Looks simple. Can be enriched by highlighting the major contributions and novelty of the study. Also, mention the methodology used. Refine the keywords carefully.

Abstract includes a brief description of the problem, the need for bioeconomy-based solutions within this field, describing the basis of the statements. The most important keywords of this review have been added to the manuscript.

  1. Introduction: Many redundant data are given. Emphasis on the important things. Carefully narrow down the research problem. More details needed on figure 1.

This is a review article, not a research article. The introduction of this review article includes the legislative framework of the reviewed topic, the circular bioeconomy principles considering forestry side-streams as the biomass in question, introduction of the topic and its topicality within the Northern Europe, and a general proposal of the necessity of circular economy principle introduction in forestry side-stream utilisation. The data reviewed illustrates the necessity for further, new studies within the field to support the introduction and development of circular solutions within forestry. Specific literature sources are an indication of the lack of R&D within this field. Figure 1 describes the proposed value pyramid of coniferous tree biomass utilisation as described in references 28,29.

  1. Chemical diversity of coniferous needles and wood greenery: Very lengthy. Explain the terms in simple words. So that it is easily understandable for everyone.

Parts in this section have been written so that the whole diversity of coniferous chemical composition is covered, the paragraphs for the various chemical groups are concise and describes the main components, including their function. This review article is written for a specific field of studies – coniferous tree biomass and its extracts, therefore it is not necessary to simplify the language or the terms for this article to be a popular science article. Moreover, most of the sections do not include any specific terms, rather general, professional terms that are used within the field of studies.

  1. Biorefinery approaches for the extraction of coniferous needles and wood greenery: Consider the length of each paragraph.

The paragraphs throughout the manuscript have been modified and changed accordingly, so that they are mor concise.

  1. Further development of coniferous needle and wood greenery biorefinery: This section can be renamed as conclusion.

This section focuses more on the future developments and the possibilities of circular economy principles within the forestry sector and more specifically the coniferous tree residual biomass, rather than concluding on the topic as it is. The future aspects in this manuscript can be understood as the conclusion of this review article, however, by changing the section headline the dimensions of the final section would be narrowed greatly.

  1. Reference: Although 147 references are provided, many are old. Consider citing recent article preferably published within last 5 years. Try to cite latest articles. The following articles will help in improving the quality of the manuscript.

The most recent articles within this specific topic have been included in the review. An extensive literature analysis has been done, which illustrates the necessity for further research within this field of bioeconomy, considering the large amount of biomass available in Northern Europe and other regions.

Thank you for the suggested articles.

Impact of Circular Bioeconomy on Industry’s Sustainable Performance: A Critical Literature Review and Future Research Directions Analysis

Inhibitors to circular economy practices in the leather industry using an integrated approach: Implications for sustainable development goals in emerging economies

Overall, the manuscript has many shortcomings. There are many inconsistencies, typo errors, and grammatical errors. Give attention to uniformity. Attention must be given to presentation of the manuscript also. Hence, I recommend major revision.

Thank you for the comments and suggestions, English of the manuscript has been improved and revised.

Yours sincerely,

Linards Klavins and the authors

Reviewer 3 Report

This review summarized the diverse chemicals in coniferous needles and wood greenery, and then listed the biorefinery methods for extracting value-added products from coniferous needles and wood greenery. The following questions should be solved.

1. Only extraction methods were mentioned in section 3, which should be pretreatment step, not a concept of biorefinery. Biorefinery usually refers to the integrated technologies to obtain pure products.

2. Coniferous needle essential oils in section 3 should be better placed in section 2.

3. Some contents in "Preparation and properties of aqueous extracts" of section 3 are similar with those in "Water soluble components of coniferous needles/greenery" of section 2.  

The initial letters of some phrases in the text should be lower-case.

Author Response

REVIEWER 3

Dear Reviewer 1,

On behalf of the authors of this review I would like to thank you for the comments that you have expressed. I believe that these implementations will improve the overall quality of the manuscript.

Please see the answers to the comments below -

This review summarized the diverse chemicals in coniferous needles and wood greenery, and then listed the biorefinery methods for extracting value-added products from coniferous needles and wood greenery. The following questions should be solved.

  1. Only extraction methods were mentioned in section 3, which should be pretreatment step, not a concept of biorefinery. Biorefinery usually refers to the integrated technologies to obtain pure products.

On Figure 1 and corresponding text on page 4 we are explaining our intents and limitations in respect to perspective of analysis of the article: our aim is to look on high added value chemicals, products for use as food and feed – the top-level products accordingly to biomass value pyramid and biorefinery concept. The use of biomass side streams as fuel, for energy production is a topic of several recent studies. Thus our intent was to analyse approaches to obtain added-value products and for these purposes extraction is the main technology and from this perspective the analysis has been done. At the same time in final parts of the study we stress the need and possibilities to develop zero-waste technologies.

  1. Coniferous needle essential oils in section 3 should be better placed in section 2.

Section 2 describes the chemical diversity, while section 3 describes the contents, methods of extraction and activity of essential oils from coniferous trees, therefore this chapter is placed in section 3.

  1. Some contents in "Preparation and properties of aqueous extracts" of section 3 are similar with those in "Water soluble components of coniferous needles/greenery" of section 2. 

Similarly to previous comment, section 3 summarizes the most commonly used methods for the preparation and uses of coniferous tree aqueous extracts, while section 2 summarizes the chemical compounds within the extract and coniferous biomass in general.

Yours sincerely,

Linards Klavins and the authors

Round 2

Reviewer 2 Report

Impact of Circular Bioeconomy on Industry’s Sustainable Performance: A Critical Literature Review and Future Research Directions Analysis Inhibitors to circular economy practices in the leather industry using an integrated approach: Implications for sustainable development goals in emerging economies

The above articles could be cited. 

Author Response

Dear Reviewer 2,

Suggestion: The above articles could be cited. 

Answer: The suggested reference has been included in the list of references and citated

Karuppiah, K., Sankaranarayanan, B., Ali, S. M., Jabbour, C. J. C., Bhalaji, R. K. A. (2021). Inhibitors to circular economy practices in the leather industry using an integrated approach: Implications for sustainable development goals in emerging economies. Sustainable Production and Consumption27, 1554-1568. https://doi.org/10.1016/j.spc.2021.03.015

Reviewer 3 Report

Most of comments are not solved well.

Author Response

Dear Reviewer 3,

On behalf of the authors of this review I would like to thank you for the comments that you have expressed. The structure of the review manuscript has been discussed with the co-authors and we believe that retaining the current structure of the article is crucial for understandability and overall readability of the article. By mixing the sections that refer to the chemical composition and used extraction methods we would somewhat confuse the reader. While the different sections contain information on the same chemical compounds/compound groups, it is important to divide the chemical diversity of coniferous extracts from the used extraction methods.

Suggestion: Only extraction methods were mentioned in section 3, which should be pretreatment step, not a concept of biorefinery. Biorefinery usually refers to the integrated technologies to obtain pure products.

Answer: On Figure 1 and corresponding text on page 4 we are explaining our intents and limitations in respect to perspective of analysis of the article: our aim is to look on high added value chemicals, products for use as food and feed – the top-level products accordingly to biomass value pyramid and biorefinery concept. The use of biomass side streams as fuel, for energy production is a topic of several recent studies. Thus our intent was to analyse approaches to obtain added-value products and for these purposes extraction is the main technology and from this perspective the analysis has been done. At the same time in final parts of the study we stress the need and possibilities to develop zero-waste technologies. To pinpoint the logical changes corresponding changes in the text of the article has been done.

Suggestion: Coniferous needle essential oils in section 3 should be better placed in section 2.

Answer: Section 2 describes the chemical diversity, while section 3 describes the contents, methods of extraction and activity of essential oils from coniferous trees, therefore this chapter is placed in section 3. Wa are absolutely sure in the suggested structuring approach of the article and believe it will be understood by the readers.

Suggestion: Some contents in "Preparation and properties of aqueous extracts" of section 3 are similar with those in "Water soluble components of coniferous needles/greenery" of section 2.

Answer: Similarly to previous comment, section 3 summarizes the most commonly used methods for the preparation and uses of coniferous tree aqueous extracts, while section 2 summarizes the chemical compounds within the extract and coniferous biomass in general.